# Learning Achievement Structure for Structured Exploration in Domains with Sparse Reward

**Zihan Zhou**
University of Toronto
Vector Institute
`footoredo@gmail.com`

**Animesh Garg**
University of Toronto
Vector Institute
`garg@cs.toronto.edu`

## Abstract

We propose *Structured Exploration with Achievements* (SEA), a multi-stage reinforcement learning algorithm designed for *achievement-based environments*, a particular type of environment with an internal achievement set. SEA first uses offline data to learn a representation of the known achievements with a determinant loss function, then recovers the dependency graph of the learned achievements with a heuristic algorithm, and finally interacts with the environment online to learn policies that master known achievements and explore new ones with a controller built with the recovered dependency graph. We empirically demonstrate that SEA can recover the achievement structure accurately and improve exploration in hard domains such as *Crafter* that are procedurally generated with high-dimensional observations like images.

## 1 Introduction

Exploration in complex environments with long horizon and high-dimensional input such as images has always been challenging in reinforcement learning. In recent years, multiple works (Stadie et al., 2015; Bellemare et al., 2016; Tang et al., 2017; Pathak et al., 2017; Burda et al., 2019a;b; Badia et al., 2020) propose to use intrinsic motivation to encourage the agent to explore less-frequently visited states or transitions and gain success in some hard exploration tasks in RL. Go-explore (Ecoffet et al., 2021) proposes to solve the hard exploration problem by building a state archive and training worker policies to reach the old states and explore new states. The shared trait of these two streams of RL exploration algorithms is that they are both built on the idea of increasing the visiting frequency of unfamiliar states, either by providing intrinsic motivation or doing explicit planning. However, This idea can be ineffective in procedurally generated environments since states in such environments can be infinite and scattered, to the extent that the same state may never appear twice. This is why exploration in such environments can be extremely hard. However, in most procedurally generated environments, there will be a finite underlying structure fixed in all environment instantiations. This underlying structure is essentially what makes the environment "meaningful". Therefore, discovering this structure becomes vital in solving procedurally generated environments.

In this work, we set our sight on *achievement-based environments*. Specifically, an achievement-based environment has a finite set of achievements. The agent only gets rewarded when it completes one of the achievements for the first time in an episode. Achievement-based environments are commonly seen in video games such as Minecraft. We argue that this type of environment also models many real-life tasks. For example, when we try to learn a new dish, the recipe usually decomposes the whole dish into multiple intermediate steps, where completing each step can be seen as an achievement. A good example of achievement-based environments in reinforcement learning community is *Crafter* (Hafner, 2021), a 2d open-world game with procedurally generated world maps that includes 22[1] achievements. We introduce this environment in more detail in Sec. 3.1.

Achievement-based environments bring both benefits and challenges to reinforcement learning exploration. On one hand, the achievement structure is a perfect invariant structure in a procedurally generated environment. On the other hand, the reward signal in such an environment is sparse since

---

[1]21 in our modified version.

the agent is only rewarded when an achievement is unlocked. The reward signal can even be confusing since unlocking different achievements grants the same reward.

We propose *Structured Exploration with Achievements* (SEA), a multi-stage algorithm that effectively solves exploration in achievement-based environments. SEA first tries to recover the achievement structure of the environment from collected trajectories, either from experts or other bootstrapping RL algorithms. With the recovered structure, SEA deploys a nonparametric meta-controller to learn a set of sub-policies that reach every discovered achievement and start from there to explore new ones.

We empirically evaluate SEA in *Crafter* and an achievement-based environment *TreeMaze* that we designed based on MiniGrid (Chevalier-Boisvert et al., 2018). In *Crafter*, our algorithm can complete the hard exploration achievements consistently and reach the hardest achievement (collect_diamond) in the game at a non-negligible frequency. None of our tested baselines can even complete any of the hard exploration achievements, and to the best of our knowledge, SEA is the first algorithm to complete this challenge. We also show that SEA can accurately recover the achievement dependency structure in both environments.

Although uncommon in the current RL scene, we argue that the achievement-based reward system can be a good environment design choice. Designing a set of achievements can be easier than crafting specific rewards system for the agent to learn. With the structure recovery module in SEA, the agent can automatically discover the optimal progression route to the desired task, alleviating the need for a high-quality achievement set. Furthermore, as shown in the *TreeMaze* experiments, the achievements can even be not semantically relevant to the tasks.

## 2 RELATED WORK

**Exploration in reinforcement learning.** Exploration in RL has long been an important topic, which can date back to $\epsilon$-greedy and UCB algorithms in multi-armed-bandit problems. In recent years, intrinsic motivation-based exploration algorithms (Stadie et al., 2015; Bellemare et al., 2016; Tang et al., 2017; Pathak et al., 2017; Burda et al., 2019a;b; Badia et al., 2020) gained great success in this area. These algorithms aim to find an estimation of the state or transition visiting frequency and use an intrinsic reward to encourage the agent to visit the less frequently visited states. However, they can be ineffective when the environment is procedurally generated, where the number of states and transitions can be infinite and scattered.

Another line of work in this area relevant to ours is the go-explore (Ecoffet et al., 2021) series. Go-explore builds an archive of explored states which are grouped into cells and learns a set of goal-conditioned policies that reach each cell. It then starts exploration from the archived cells to find new cells and updating the archive iteratively. Go-explore shows success in the hard exploration tasks in Atari games. However, go-explore can also be ineffective in procedurally generated environments since it uses image down-sampling as the default state abstraction method. Another point where our work differs from go-explore is that we don't seek to group all states into cells, but instead only group the few import ones. This is helpful since archiving all the states and learning all the goal-conditioned policies can take up huge amount of resources.

**Hierarchical reinforcement learning.** Hierarchical reinforcement learning (HRL) explores the idea of dividing a large and complex task into smaller sub-tasks with reusable skills (Dayan & Hinton, 1992; Sutton et al., 1999; Bacon et al., 2017; Vezhnevets et al., 2017; Nachum et al., 2018). Eysenbach et al. (2019) and Hartikainen et al. (2020) propose to first learn a set of unsupervised skills then build a policy with the learned skills for faster learning and better generalization; Bacon et al. (2017) builds a set of policy options to learn. Goal-conditioned HRL (Vezhnevets et al., 2017; Nachum et al., 2018; 2019) train a manager policy that proposes parameterized goals and a goal-conditioned worker policy to reach the proposed goals. Kulkarni et al. (2016); Lyu et al. (2019); Rafati & Noelle (2019); Sohn et al. (2020); Costales et al. (2022) train a set of sub-policies to solve the subtasks provided by the environment and a meta-controller to maximize the reward. This line of work is similar to ours in that they assume the subtasks are provided by the environment. However, in our work, we don't assume access to any additional subgoal information such as subgoal completion signals from the environment that are typically required in this line of work. Additionally, the

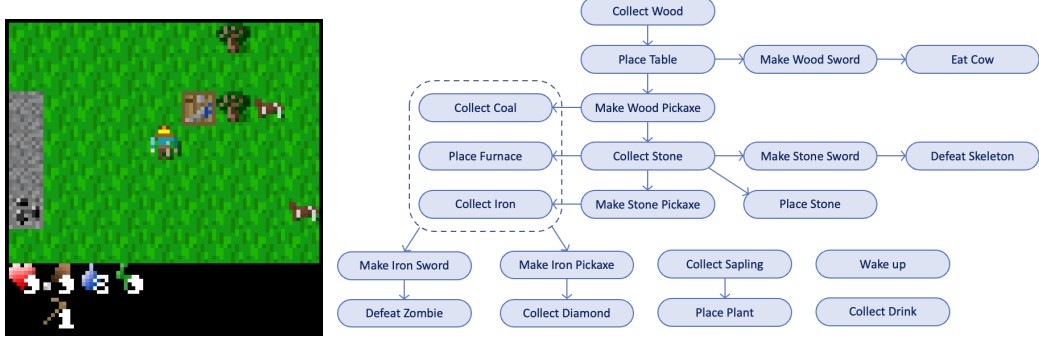

(a) Crafter environment.

(b) Crafter achievements dependency graph.

Figure 1: The Crafter environment.

problems that we are solving are more complex, namely including more tasks and being image-based and procedurally generated.

Despite being conceptually similar to the subgoal/subtask in HRL, the achievement in our work bears a different meaning and serves a different purpose. For an extended discussion on this topic, the reader can refer to Section D.1 in the appendix.

**Building world models.** Watter et al. (2015); Karl et al. (2017); Ha & Schmidhuber (2018); Hafner et al. (2021) argue that learning a compact latent space world model is beneficial in environments with high-dimensional inputs such as images. $L^3P$ (Zhang et al., 2021) proposes to represent the world model as a graph by clustering states into nodes, similar to SEA. However, $L^3P$ learns the state representation with the objective to recover reachability, which is not guaranteed to be useful in exploration.

**Reward machines.** Reward machines (Icarte et al., 2018a;b) is a concept that uses linear temporal logic language to describe a non-Markovian reward function. Recent works (Camacho et al., 2019; Icarte et al., 2019; 2022) exploit this concept to do structured reinforcement learning and improve performance. However, they depend on the environment providing a labeling function to learn the reward machine instead of learning one from the environment. Achievement-based environments also feature a non-Markovian reward function since completing one achievement only gets rewarded once in an episode, therefore it can be seen as a special instance of the reward machines.

## 3 PROBLEM SETUP

### 3.1 CRAFTER

To better explain the achievement-based environment, we introduce *Crafter*[2] (Hafner (2021), Fig. 1a). Crafter is a 2d open-world game featuring a procedurally generated world map and 21 in-game achievements. In this game, the agent needs to control the character to explore the world by collecting materials, crafting tools, and defeating monsters. Whenever the agent completes a certain achievement for the first time in an episode, it will receive a reward of one. For any subsequent completion, the agent will not be rewarded. The goal of the game is to unlock as many achievements as possible. In Crafter, some achievements may have dependencies on others. For example, the make_wood_pickaxe achievement depends on the place_table achievement since crafting a wood pickaxe requires the agent to be near a crafting table. These achievements and their dependencies form a directed acyclic graph (DAG), demonstrated in Fig. 1b.

### 3.2 MARKOV DECISION PROCESS WITH ACHIEVEMENTS

We formalize our definition of achievement-based environments as a *Markov decision process with achievements* (MDPA). A Markov decision process with achievements is defined by a 5-tuple

---

[2]We use a slightly modified version of Crafter. The detailed modification can be found in Appendix.

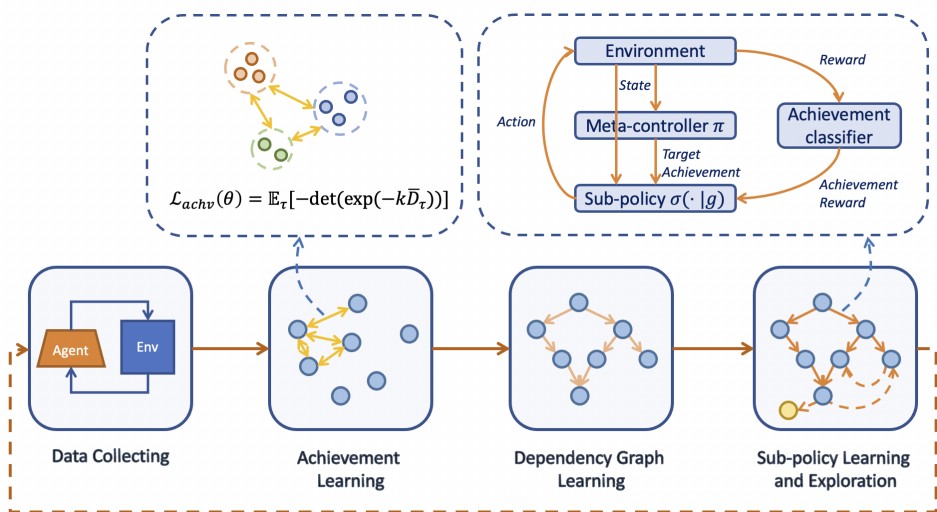

Figure 2: Algorithm overview. SEA comprises of four parts. (1) It starts with collecting trajectories either from domain experts or previously trained agents. (2) With the collected trajectories, SEA learns the achievement representation with Eqn. 1 and builds an achievement classifier by clustering the learned representations. (3) SEA then recovers an achievement dependency graph with the classifier and the collected trajectories. (4) We train a set of sub-policies to reach each recognized achievement and an exploration policy for new achievements. A meta-controller is used to propose target achievements according to the achievement dependency graph. We use the achievement classifier to filter out the reward signal for the target achievement.

$(\mathcal{S}, \mathcal{A}, T, \Gamma, G)$, where $\mathcal{S}$ is the set of states, $\mathcal{A}$ is the set of actions, $T : \mathcal{S} \times \mathcal{A} \rightarrow \Delta^{\mathcal{S}}$ is the stochastic state transition function, $\Gamma$ is the set of achievements, and $G : \mathcal{S} \times \mathcal{A} \times \mathcal{S} \rightarrow \Gamma \cup \{\emptyset\}$ is the achievement completion function. For any state transition tuple $(s_t, a_t, s_{t+1})$, the achievement completion function maps it to either one of the achievements in $\Gamma$, meaning that particular achievement is completed at this step, or $\emptyset$, meaning none of the achievements is completed. Achievement unlocking reward can be derived from an MDPA trajectory. Specifically, for a trajectory $\tau = (s_0, a_0, s_1 \ldots, s_T)$, the reward at step $t \in 0, \ldots, T - 1$ is:

$$r_t := \mathbb{1}[G(s_t, a_t, s_{t+1}) \neq \emptyset] \prod_{t'=0}^{t-1} \mathbb{1}[G(s_t, a_t, s_{t+1}) \neq G(s_{t'}, a_{t'}, s_{t'+1})],$$

where $\mathbb{1}[\cdot]$ is the indicator function. Note that directly adding the reward function into an MDPA breaks its Markovian property since the agent will only be rewarded for the first time it completes an achievement. We can instead derive a standard MDP from an MDPA by defining a set of achievement-extended states $\mathcal{S}^* = \mathcal{S} \times 2^{\Gamma}$ where we store the achievement completion information into the state. We can also define the corresponding state transition function $T^* : \mathcal{S}^* \times \mathcal{A} \rightarrow \mathcal{S}^*$ and reward function $R : \mathcal{S}^* \times \mathcal{A} \times \mathcal{S}^* \rightarrow \mathbb{R}$ to complete the derivation.

## 4 METHOD

The advantage of learning in MDPA is that achievements, hinted by reward signals, are invariant across different environment instantiations. These achievements can therefore be seen as checkpoints, which allow us to do a divide-and-conquer of the hard environment exploration task.

Following this argument, we propose an algorithm *Structured Exploration with Achievements* (SEA), which recovers the underlying achievement structure and conducts structural explorations. Specifically, we use the data collected from either previously trained agents or experts to recover the known achievements and their corresponding structures. With the recovered achievement structure, we then train multiple sub-policies to reach every discovered achievement and start individual searches for new achievements from there.

## 4.1 ACHIEVEMENT LEARNING

The first step of our method is to identify the achievements in the environment by learning the representations of the achievements and then clustering the achievements with their representations. For any trajectory $\tau = (s_0, a_0, \ldots, s_T)$, we define $U_\tau := \{t : r_t = 1\}$ be the step set where an achievement is unlocked. For a representation dimension $W$, we want to train a model parameterized by $\theta$ that maps the tuple $(s_t, a_t, s_{t+1})$ where $t \in U_\tau$, to a vector of size $W$. We achieve this by using the uniqueness property of achievements, that every two steps in $U_\tau$ correspond to different achievements. In other words, we want to ensure the representations of each pair of steps in $U_\tau$ are as distinct as possible.

Let $\bar{s}_t := (s_t, a_t, s_{t+1})$. We define the achievement representation distance matrix of trajectory $\tau$ as $D_\tau := [\|\phi_\theta(\bar{s}_{t_1}), \phi_\theta(\bar{s}_{t_2})\|_2]_{t_1, t_2 \in U_\tau}$ and $\overline{D}_\tau := D_\tau / \max D_\tau$.

Normalization is used to prevent the representation from outward expanding, since expanding increases the pairwise distance but not the separability required for clustering. To maximize the pairwise distance of the achievement representations, we use the following loss:

$$\mathcal{L}_{\text{achv}}(\theta) := \mathbb{E}_\tau \left[ -\det \left( \exp \left( -k\overline{D}_\tau \right) \right) \right], \tag{1}$$

where $\det(\cdot)$ calculates the determinant of a matrix. The loss reaches the minimum value of -1 when every pair of achievement representations in a trajectory are sufficiently different so that the matrix is close to the identity matrix. This formulation is inspired by Kulesza (2012) and Parker-Holder et al. (2020). Compared to directly maximizing the average pairwise distances, the advantage of this formulation is that it prevents the different achievement representations in an episode from forming clusters. In such case, the average pairwise distance is still high but the determinant loss is close to zero.

We then use the K-Means algorithm for clustering. Suppose we cluster the achievements into $N$ clusters with centroids denoted as $c_1, \ldots, c_N$. We can now build an achievement classifier $AC(\bar{s}) := \arg\min_i \|\phi_\theta(\bar{s}) - c_i\|_2$. However, to improve the generalization of this classifier, we would like it to gain the ability to recognize new achievements that are not in the dataset. To do this, we set a threshold $b = 0.5 \cdot \min_{i,j} \|c_i - c_j\|_2$. Let $i^* = \arg\min_i \|\phi_\theta(\bar{s}) - c_i\|_2$, the new classifier becomes:

$$AC(\bar{s}) := \begin{cases} i^* & \text{if } \|\phi_\theta(\bar{s}) - c_{i^*}\|_2 < b, \\ -1 & \text{otherwise}, \end{cases}$$

where $-1$ means $\bar{s}$ is a unrecognized new achievement.

## 4.2 ACHIEVEMENT DEPENDENCY GRAPH

With the achievements identified, we then use a heuristic algorithm to build a preliminary achievement dependency graph. The algorithm comprises of two parts: 1) we build a complete dependency graph; 2) we trim down the complete graph into a minimal DAG.

We represent both graphs with an adjacency matrix of $N$ nodes. We first calculate two heuristics of the collected trajectories.

$$\text{BEFORE}_{ij} := |\{v_{t_1} = i \ \wedge \ v_{t_2} = j \mid t_1 < t_2 \in U_\tau, \ \forall \tau\}|$$
$$\text{HAPPEN}_i := |\{v_t = i \mid t \in U_\tau, \ \forall \tau\}|$$

We say $i$ is a dependency of $j$ if every time $j$ happens, $i$ happens before $j$. Therefore, for the first part,

$$G_{ij} := \mathbb{1}[\text{BEFORE}_{ij}/\text{HAPPEN}_j > 1 - \epsilon],$$

where $\epsilon$ is to account for identification error. For a small $\epsilon$, $G$ is guaranteed to be a DAG.

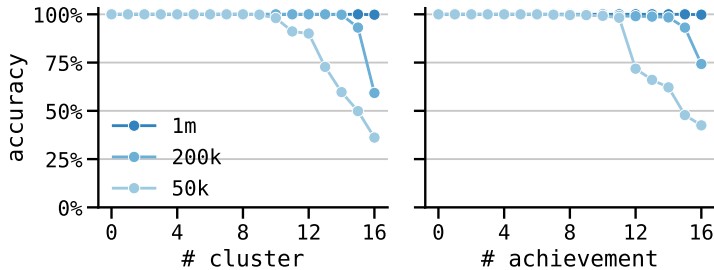

Figure 3: Clustering accuracy with different data sizes. Clusters and achievements are sorted in descending order of prediction accuracy.

For the second part, we need to remove the unnecessary edges in $G$ to get a minimal DAG while maintaining the topology of $G$. In this case, an edge is considered unnecessary if removing it does not affect the topological order of the graph. Since removing an edge $i \rightarrow j$ can only allow node $j$ to appear earlier in the topological order, we only need to calculate the left-most position of a node, denoted as $\text{LEFTMOST}(G, i)$.

$$G_{ij}^* := G_{ij} \cdot \mathbb{1}[\text{LEFTMOST}(G \backslash \{i \rightarrow j\}, j) < \text{LEFTMOST}(G, j)].$$

We point out that our recovered graph may include more dependency edges than the ground truth owing to the non-causal correlations from either agent preference or insufficient data. We will address this issue in the next section.

### 4.3 SUB-POLICY LEARNING AND EXPLORATION

The last part of our algorithm is to learn a set of sub-policies that reach every achievement node and do exploration from them. A sub-policy is a parameterized achievement-conditioned policy $\sigma_\omega(a|s, g)$ where $\omega$ is the parameter shared among all sub-policies and $g \in \Gamma$ is the target achievement. Each sub-policy only receives a reward of 1 upon reaching its target achievement, which can be done using the achievement classifier in Sec. 4.1.

To learn the sub-policies, we also need a meta-controller $\pi(g|s)$ that proposes a target achievement to learn. One option is to randomly select a new one when an achievement is completed. However, this can be ineffective when a target achievement is proposed before all its dependencies are completed. This is because the sub-policy only has access to the reward of its target achievement, which makes the exploration much harder. Therefore, we instead use a controller that topologically traverses the dependency graph. At the beginning of the episode or upon completion of any achievement, the controller randomly selects an unfinished achievement whose dependent achievements have all been completed. In our implementation, we also bias the selection probability towards a direct child of the last completed achievement to provide continuity.

The unnecessary dependencies introduced in the last step won't affect the effectiveness of sub-policy learning since following the recovered dependency graph still guarantees the necessary topological order. To explore a more accurate graph, we let the meta-controller selects the new target randomly with a small possibility. We refer the readers to Sec. C.1.1 for more details on the meta-controller.

## 5 EXPERIMENTS

### 5.1 EXPERIMENT SETUP

Our experiments are designed to evaluate the three main parts of our algorithm: achievement learning, graph recovery, and sub-policy learning & exploration. To better investigate different aspects of our algorithm, we design a pedagogical environment based on MiniGrid (Chevalier-Boisvert et al., 2018) called *TreeMaze* (Fig. 5). *TreeMaze* has 2 columns of $N$ rooms. Each room contains either a key that unlocks other rooms, an object that can be interacted with, or nothing. A key can only unlock doors with the same color as the key, while some of the rooms are not locked. In *TreeMaze*,

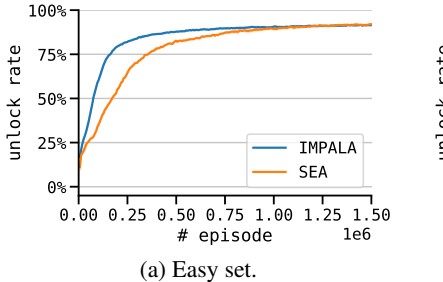
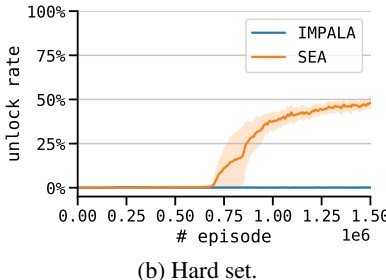

(a) Easy set.  (b) Hard set.

Figure 4: Mean unlocking rate comparison in easy and hard achievement set.

every interaction, including opening a door, picking up a key, and interacting with other objects, is seen as an achievement. The rooms are generated with the guarantee that it's possible to unlock all achievements. The locations of the rooms are randomized but the dependency graph is fixed. We use three versions of *TreeMaze* in our experiments, the *TreeMaze-Easy* which contains 4 rooms and 9 achievements, the *TreeMaze-Normal* which contains 10 rooms and 19 achievements and the *TreeMaze-Hard* that contains 18 rooms and 30 achievements.

For the *Crafter* environment, we also include a limited version (*Crafter-Limited*) which only contains 7 of the 21 achievements in the full environment.

We train a vanilla IMPALA (Espeholt et al., 2018) agent for the initial data gathering in each environment. In the full *Crafter* environment, the IMPALA agent discovers 17 achievements (with $> 1\%$ reaching rate); In all the remaining environments, the IMPALA agents discover all of the achievements. We consider the 17 achievements reachable by a vanilla IMPALA agent as the easy achievement set, and the remaining four achievements (`make_iron_pickaxe`, `make_iron_sword`, `defeat_zombie`, `collect_diamond`) as the hard achievement set. The reason why these achievements are particularly hard to

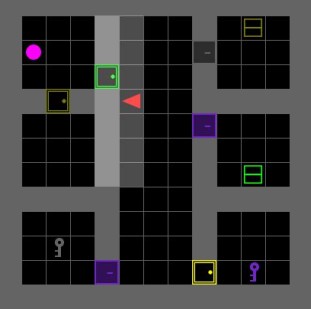

Figure 5: *TreeMaze*.

complete is that crafting iron tools in *Crafter* has three prerequisites: the character near a crafting table, the character near a furnace, the character has at least one wood, one coal, and one iron in the inventory. Of the 4 hard achievements, `collect_diamond` is objectively the hardest achievement to unlock as not only does it depends on the `make_iron_pickaxe`, but also that the diamond resource is extremely scarce in the game. The complete achievement list can be found in Sec. B.1.

## 5.2 ACHIEVEMENT LEARNING

We first investigate the clustering accuracy of our algorithm. As an evaluation metric, we calculate both cluster purity, which is defined as the proportion of the majority achievement in each cluster, and achievement purity, which is defined as the proportion of the majority cluster that the achievement is clustered into. The final clustering accuracy is defined as the mean purity across all clusters and achievements.

|  | Latent | SEA -det | SEA |
|---|---|---|---|
| *Crafter-Limited* | 0.755 | | **1.000** |
| *Crafter* | N/A | 0.966 | **1.000** |
| *TreeMaze-Easy* | 0.735 | | **1.000** |
| *TreeMaze-Normal* | N/A | | **0.994** |
| *TreeMaze-Hard* | N/A | N/A | **0.996** |

Table 1: Clustering accuracies.

**Baseline and result.** We design a latent space-based baseline for achievement learning. Specifically, we train a neural network to predict the environmental reward and use the latent variable of the network as the achievement representation. The idea behind this baseline is that to predict a certain achievement being completed, only the necessary information in the observation will be preserved in the predictor network. To test the effectiveness of our determinant-based loss function, we add an ablation experiment that replaces the determinant in Eqn. 1 with mean pairwise distances. The clustering accuracy can be found in Tab. 1. The latent-space baseline only manages to provide meaningful clustering in two of the easiest environments, while our algorithm does almost-perfect clustering in all tested environments. The mean distance variant shows slightly worse clustering

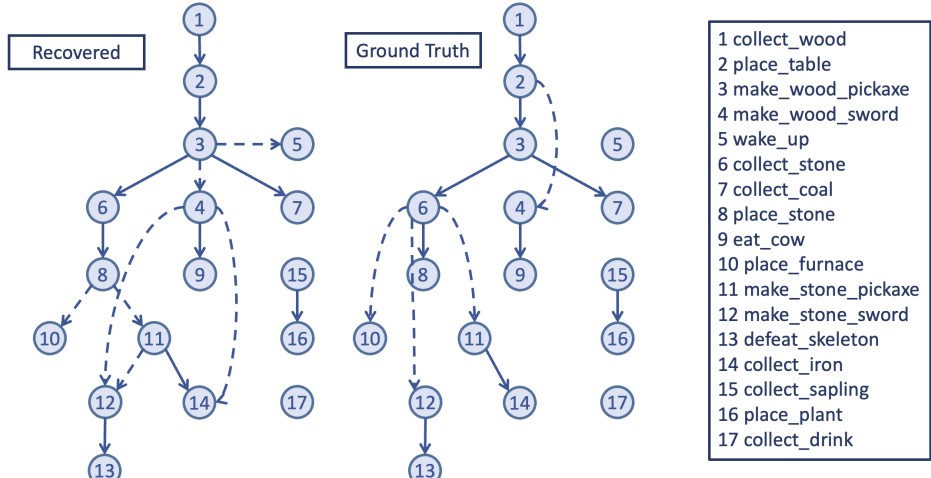

Figure 6: Recovered dependency graph comparison. Dashed lines represent differences.

performance than SEA in the *Crafter* environment but completely fails in the *TreeMaze-Hard* environment, which has the largest amount of achievements.

We note here that even though the determinant loss massively outperforms the latent-space baseline in our tested environments, it has the advantage of not being dependent on the achievement reward assumption. In our preliminary testing, we find out that the latent-space baseline separates different sources of reward better in some environments where the achievement reward assumption does not hold.

**Sample efficient.** We further investigate the sample efficiency of our algorithm. We run three experiments with 1 million, 200 thousand, and 50 thousand environment steps of data, respectively. The clustering accuracy in Crafter is shown in Fig. 3. Numerically, 1 million steps of data are sufficient as it reaches $> 99\%$ accuracy in all clusters and achievements, while only using 200 thousand steps also reaches $> 90\%$ accuracy in 16 clusters/achievements out of all 17 ones.

## 5.3 STRUCTURE RECOVERY

In all three *TreeMaze* environments and *Crafter-Limited*, our algorithm successfully recovers the exact dependency graphs. In full *Crafter*, the graph Hamming distance (defined as the number of different entries in the adjacency matrix) between our recovered graph and the ground truth graph is 11 out of 272 entries. The detailed graph comparison can be found in Fig. 6.

## 5.4 SUB-POLICY LEARNING AND EXPLORATION

**Meta-controller comparison.** We first show the effectiveness of the dependency graph-based meta-controller in training the sub-policies. The mean easy achievement set unlocking rate compared to a random controller is shown in Fig. 7. As shown in the figure, a graph-based meta-controller learns the sub-policies much faster than a random controller.

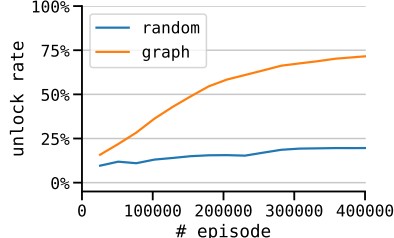

Figure 7: Comparison of mean achievement unlocking rate with a graph-based meta-controller and a random one.

**Sub-policy learning and exploration.** For this part, we choose IMPALA (Espeholt et al., 2018) and PPO (Schulman et al., 2017) as widely-recognized model-free baselines, IMPALA + RND (Burda et al., 2019b) as an exploration baseline, DreamerV2 (Hafner et al., 2021) as the model-based baseline since it achieves the highest reward in the original *Crafter* paper (Hafner, 2021), and HAL[3] (Costales et al., 2022) as a Hierarchical reinforcement learning baseline.

_________

[3]Note that HAL has an advantage over the other selected algorithms since it requires the environment to provide extra information of the achievements, such as achievements completion and affordance signals.

To evaluate SEA's ability to master existing tasks and explore new tasks separately, we compare the mean achievement unlock rate[4] in the easy achievement set and the hard achievement set respectively, as shown in Fig. 4. From Fig. 4a, we notice that SEA learns slightly slower than vanilla IMPALA in the easy set. This is because SEA is not designed to maximize the total reward but to master each achievement-targeted sub-policy, as our meta-controller does not try to learn an optimal achievement completion route. From Fig. 4a, we can see that SEA successfully discovers the hard achievements with a significant reaching rate while vanilla IMPALA fails to reach any of them, which shows the exploration potential of our algorithm. The complete unlocking rate curve can be found in Fig. 8 in the Appendix.

A detailed numerical comparison with all baselines is shown in Table 2 and 3. We observe that SEA is the only algorithm that can consistently reach the hard set achievements and at the same time maintains close-to-optimum easy set achievements reaching rate. SEA is also able to reach the hardest achievement in *Crafter*, `collect_diamond`, with a non-negligible probability.

Table 2: Achievement unlock rate comparison. Standard deviation shown in the parenthesis.

| | *Easy Set* | | *Hard Set* | | |
| | mean | median | mean | median | `collect_diamond` |
| --- | --- | --- | --- | --- | --- |
| IMPALA | **93.91%** (0.12%) | **99.25%** (0.35%) | 0.00% (0.00%) | 0.00% (0.00%) | 0.00% (0.00%) |
| RND | 93.24% (1.08%) | 98.58% (0.59%) | 0.15% (0.21%) | 0.13% (0.18%) | 0.00% (0.00%) |
| PPO | 34.86% (2.19%) | 21.13% (5.48%) | 0.00% (0.00%) | 0.00% (0.00%) | 0.00% (0.00%) |
| DreamerV2 | 76.21% (11.19%) | 92.00% (1.41%) | 0.00% (0.00%) | 0.00% (0.00%) | 0.00% (0.00%) |
| HAL | 15.76% (0.75%) | 0.60% (0.57%) | 0.00% (0.00%) | 0.00% (0.00%) | 0.00% (0.00%) |
| SEA | 92.53% (0.54%) | 96.84% (0.58%) | **49.30%** (2.85%) | **60.70%** (1.67%) | **4.21%** (3.24%) |

Table 3: Crafter score[5]. Standard deviation shown in the parenthesis.

| IMPALA | RND | PPO | DreamerV2 | HAL | SEA |
| --- | --- | --- | --- | --- | --- |
| 38.54 (0.04) | 39.21 (0.91) | 6.31 (1.02) | 24.79 (9.78) | 2.41 (0.18) | **75.52** (2.36) |

**Sample efficiency.** For SEA, we use 200 million environment steps to train the original IMPALA policy for the initial data collecting and 300 million steps for the sub-policy learning and exploration. For both vanilla IMPALA and IMPALA + RND, we train the agent for 1 billion steps. For PPO, we only manage to run the agent for 110 million steps due to the time limit. For DreamerV2, we run the agent for 2.5 million steps. For HAL, we run the agent for 140 million steps. We also note that for PPO, DreamerV2, and HAL, even though the environment steps do not match SEA, their training wall clock times are all longer than the 1 billion step IMPALA experiment and we don't observe significant improvement for at least the last 1/3 of the training. We refer the reader to Fig. 8 in the Appendix.

## 6 CONCLUSION AND DISCUSSION

Exploration in procedurally generated environments is particularly hard since state frequency-based exploration methods are often not effective in such domains. We set our sights on achievement-based environments and propose *Structured Exploration with Achievements* (SEA), a multi-stage algorithm specialized in exploring in such domains. Empirically, SEA successfully recovers the underlying structure of *Crafter*, a procedurally generated achievement-based environment, and uses the recovered structure to complete the hard exploration tasks in *Crafter*. We notice that achievement-based environments are not common in the reinforcement learning community. However, since achievement-based environments are easy to design and can help to learn hard exploration tasks even without a particularly good choice of achievement set, we call on the community to consider this design choice more often when developing new environments.

---

[4]The unlock rate refers to the frequency of unlocking a certain achievement.

[5]We follow the definition from Hafner (2021): $S := \exp(\frac{1}{N} \sum_{i=1}^{n} \ln(1 + s_i)) - 1$, where $s_i \in [0, 100]$ is the percentage unlock rate for each achievement.

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

# A ADDITIONAL EXPERIMENT RESULTS

## A.1 CRAFTER INDIVIDUAL ACHIEVEMENT LEARNING CURVE

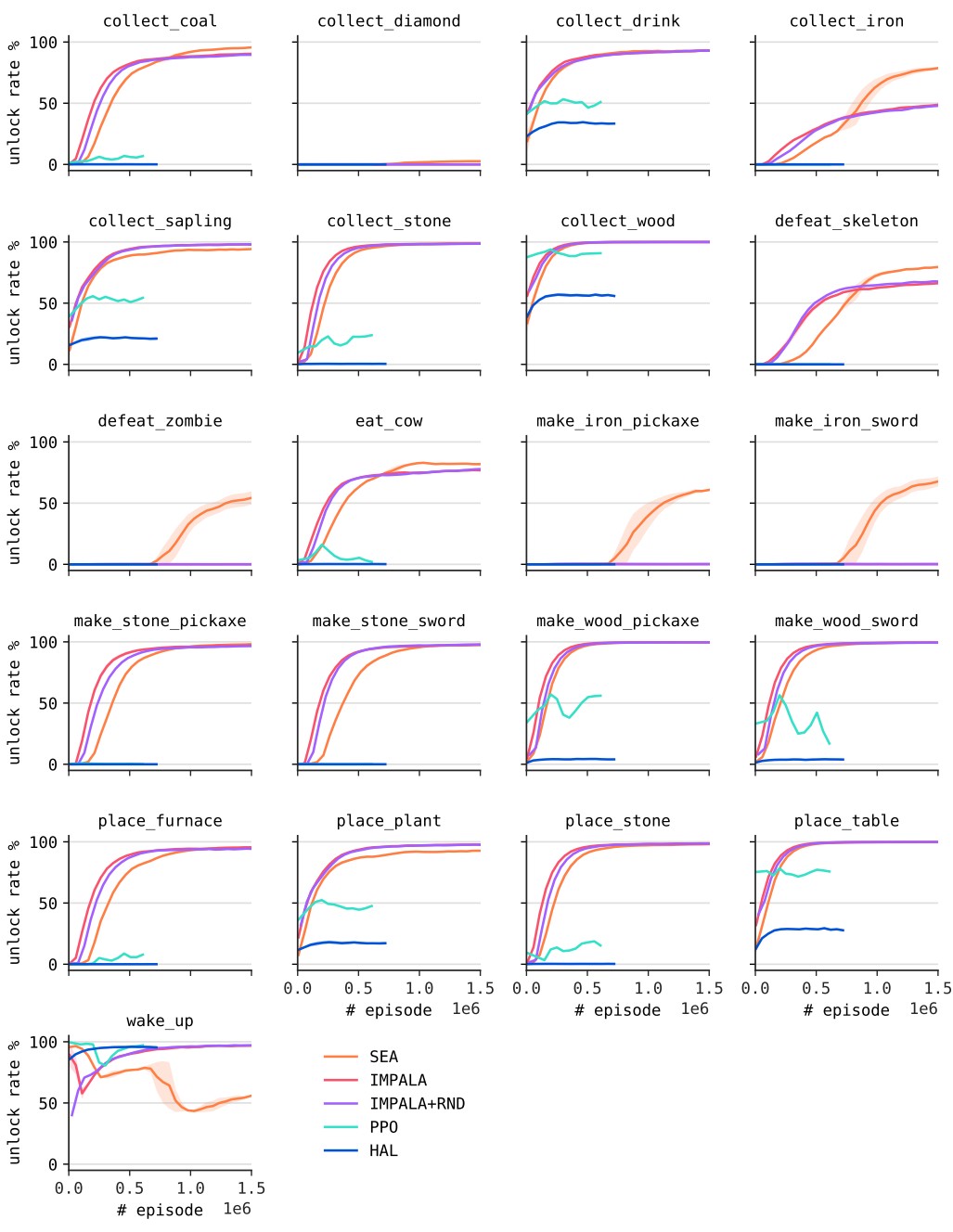

Figure 8: Achievement unlock rate curve for each individual task in Crafter, excluding DreamerV2.

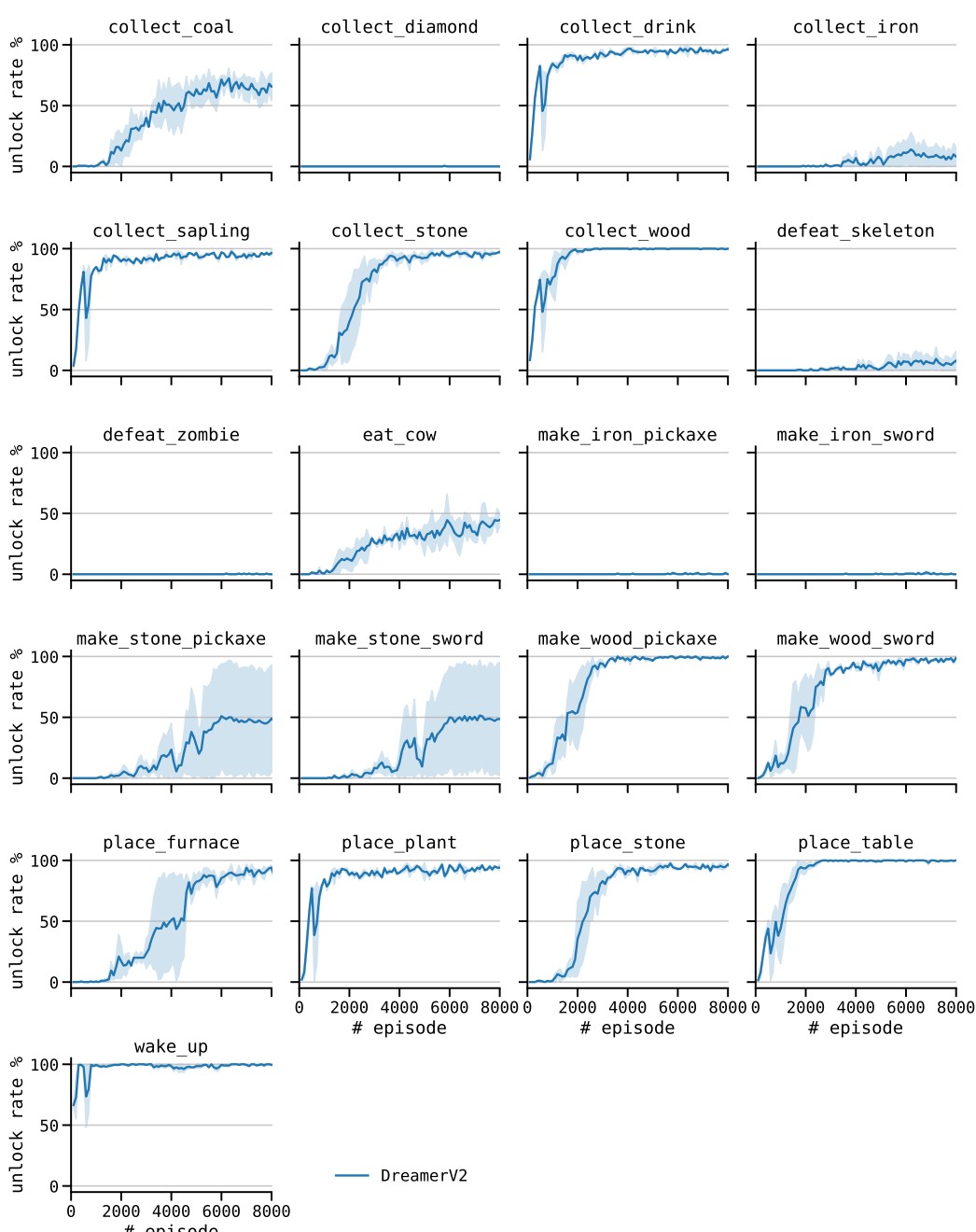

Figure 9: Achievement unlock rate curve for each individual task in Crafter, DreamerV2.

# B  ENVIRONMENT DETAILS

## B.1  COMPLETE CRAFTER ACHIEVEMENT LIST

The complete list of Crafter achievements are {collect_coal, collect_diamond, collect_drink, collect_iron, collect_sapling, collect_stone, collect_wood, defeat_skeleton, defeat_zombie, eat_cow, make_iron_pickaxe, make_iron_sword, make_stone_pickaxe, make_stone_sword, make_wood_pickaxe, make_wood_sword, place_furnace, place_plant, place_stone, place_table,

wake_up}. Of those the 17 achievements {collect_coal, collect_drink, collect_iron, collect_sapling, collect_stone, collect_wood, defeat_skeleton, eat_cow, eat_plant, make_stone_pickaxe, make_stone_sword, make_wood_pickaxe, make_wood_sword, place_furnace, place_plant, place_stone, place_table, wake_up} are the easy set since they are solvable by the vanilla IMPALA agent. The remaining 4 tasks {collect_diamond, defeat_zombie, make_iron_pickaxe, make_iron_sword} are the hard set.

## B.2 MODIFICATIONS TO THE ORIGINAL CRAFTER ENVIRONMENT

We make some modifications to the original Crafter environment to make it better suits our setting. The complete change list is:

- The health reward is removed since it does not fit into our achievement reward assumption. Consequently, the character won't lose health point in any way.

- The game ends if no achievement has been unlocked in the last 100 steps.

- When a creature (including a cow, a zombie and a skeleton) gets attacked by the character, it will not lose health point unless the attacking point of the character is greater than or equal to the health point of that creature. With this modification, the eat_cow, defeat_skeleton and defeat_zombie achievements can be connected to the main dependency graph instead of being independent nodes.

## C REPRODUCEBILITY

### C.1 ADDITIONAL ALGORITHM DETAILS

#### C.1.1 META-CONTROLLER DETAILS

Let the dependency graph be $G$. The meta-controller maintains a set of the completed achievements $S$ that are initialized to $\emptyset$. At the beginning of an episode and whenever an achievement has been unlocked, the meta-controller (re)generates a new target achievement.

To generate a new target achievement, if it is the beginning of an episode, the meta-controller randomly selects an achievement with no dependencies according to the dependency graph; Otherwise, suppose the just completed achievement is $u$. Let $V = \{v \notin S \mid w \in S, \forall w \to v \in G\}$ be the available achievements set. Let $\mathrm{CONT}(u) = \{v \in V \mid u \to v \in G\}$ be the subset of $V$ that are direct descendants of $u$. Let $\mathrm{NGHB}(u) = \{v \in V \mid \exists w \in V \text{ s.t. } v \to w \in G\}$ be the subset that are neighbours of $u$ (i.e. on which one or more $u$'s direct descendants are depended). Let $\mathrm{JUMP}(u) = \{v \in V \mid v \notin \mathrm{CONT}(u) \land v \notin \mathrm{NGHB}(u)\}$ be the remaining achievements. Notice these three subsets partitions $V$. The meta-controller stochastically chooses the target achievement according to the following probabilities:

- With probability 40%, the meta-controller chooses the exploration task;

- With probability 48%, the meta-controller uniformly randomly chooses an achievement from $\mathrm{CONT}(u)$;

- With probability 6%, the meta-controller uniformly randomly chooses an achievement from $\mathrm{NGHB}(u)$;

- With probability 6%, the meta-controller uniformly randomly chooses an achievement from $\mathrm{JUMP}(u)$.

After selecting the next target, the meta-controller updates the completed achievements set $S \leftarrow S \cup \{u\}$.

## C.2 IMPLEMENTATION

Our code is adapted from an open-source implementation of IMPALA, *torchbeast*[6]. The complete code can be found here[7].

## C.3 HYPERPARAMETERS

Table 4: IMPALA hyperparameters for different experiments.

| Hyperparameters | *Crafter* | *TreeMaze* |
|---|---|---|
| Network structure | CNN+LSTM | CNN+LSTM |
| Hidden size | 256 | 256 |
| Initial learning rate | 0.0002 | 0.0002 |
| Batch size | 32 | 32 |
| Unroll length | 80 | 80 |
| Gradient clipping | 40 | 40 |
| RMSProp $\alpha$ | 0.99 | 0.99 |
| RMSProp momentem | 0 | 0 |
| RMSProp $\epsilon$ | 0.01 | 0.01 |
| Discount rate ($\gamma$) | 0.99 | 0.99 |
| Normalize reward | Yes | Yes |

## C.4 IMPLEMENTATION OF ACHIEVEMENT REPRESENTATION FUNCTION $\phi$

$\phi$ uses the two networks to encode the states, one for the current state $s_t$, one for the next state $s_{t+1}$. The network structure is the same as the policy network. The encoding of the two states are concatenated into the observation vector $x_{obs}$. $x_{obs}$ is then concatenated with the one-hot representation of the current action $a_t$. The concatenated vector is then fed into a fully-connected layer, resulting in $x_{rep}$. The final output of $\phi$ is $x_{obs} + x_{rep}$.

# D ADDITIONAL DISCUSSIONS

## D.1 COMPARISON WITH HRL

Despite sharing similar concepts (achievements and subgoals/subtasks) with HRL, our work has a different focus. Subgoals and subtasks in HRL normally refer to the intermediate steps that lead to an ultimate goal/task. Achievements are all equal in nature, although some achievements can be the subgoals to other achievements. Our algorithm can automatically determine which achievements can be useful (potentially as subgoals) to other achievements. This lowers the standard for the achievement design as there is no need to carefully craft a series of intermediate steps that help the agent to learn. Our achievement setting also allows for multiple ultimate tasks that our algorithm can solve for in one go.

---

[6]https://github.com/facebookresearch/torchbeast
[7]https://github.com/footoredo/iclr23-sea

