# OpenReview forum: "Learning Achievement Structure for Structured Exploration in Domains with Sparse Reward"
_ICLR.cc/2023/Conference — ICLR 2023 poster_

### Official Review · Reviewer_kuVW · 2022-10-24

**Confidence:** 3
**Correctness:** 3
**Technical Novelty And Significance:** 3
**Empirical Novelty And Significance:** 3
**Recommendation:** 8

**Clarity, Quality, Novelty And Reproducibility:**

In general the paper is well written and easy to follow. While the work does make some assumptions about the underlying problems, I think the current version serves well as a starting point. In terms of novelty, I am unaware of any methods that address the same problem or provide the same level of flexibility. With that said, the authors should try to expand their related works section to contrast the current method with other approaches that leverages information from symbolic models (apart from [1], there are works like [2],[3] etc..) or works that try to generalize from information learned from simpler domain instances (for example [4]).

[2] Yang, Fangkai, et al. "Peorl: Integrating symbolic planning and hierarchical reinforcement learning for robust decision-making." arXiv preprint arXiv:1804.07779 (2018).

[3] Lee, Junkyu, et al. "AI Planning Annotation for Sample Efficient Reinforcement Learning." arXiv preprint arXiv:2203.00669 (2022).

[4] Groshev, Edward, et al. "Learning generalized reactive policies using deep neural networks." Twenty-Eighth International Conference on Automated Planning and Scheduling. 2018.

**Strength And Weaknesses:**

I mostly want to highlight two aspects, how easy it is to receive the assumed trajectories and whether the work is making more assumptions about the achievements than what is listed.

Trajectories: So one of the underlying assumptions of the method is access to trajectories that reflect the underlying achievement structure. The effectiveness of the method relies on how well these trajectories capture the underlying structure. The work mostly mentions that these trajectories could come from experts or other previously trained agents. However, this is built on the assumption that there exists a small/easy enough instance which shares the same structure, but where the exploration can be performed effectively (or where the expert can give the traces). This need not be true and the simplest problem instance may still be too hard to effectively specify all the trajectories. One possibility the authors might want to consider is, how easy would be for domain experts to directly specify, at least an incomplete version of the achievement dependency graph. This is a common practice in many fields like causal inference and learning. Also if a set of trajectories is available, a possibility that should have been considered was whether you could have used an inverse reinforcement learning algorithm to learn a potentially informative reward function. Provided these reward functions are represented in parametric form, it should be possible to apply them within other instances at least as a way to provide shaping information.

Assumptions about achievement structures: The heuristic algorithm seems to imply that the algorithm would only consider an ordering to be valid if it holds in every trajectory. While this is a reasonable starting point, this would overlook cases when there are disjunctive relationships between parents and children. A common case could be one when a specific fact could be achieved in a different way (For example, you can enter a room by either unlocking the door or by breaking down the door). Similarly the current method also seems to make the assumption that all transitions that correspond to a specific achievement would be equivalent in how good they are with respect to the ease with which future achievements can be reached from the resulting state. This is a common assumption shared by a lot of works that tries to break down the problem into subgoals and then solve subgoals independently. You can look at [1] to see an example of a recent work that tries to address this issue.

[1] Guan, Lin, Sarath Sreedharan, and Subbarao Kambhampati. "Leveraging Approximate Symbolic Models for Reinforcement Learning via Skill Diversity." ICML (2022).


**Summary Of The Paper:**

The paper looks at a specific class of RL problems where the reward structure of the underlying MDP can be specified in terms of achievements. The paper proposes a multi-stage method that tries to uncover the structure of the problem, by learning an achievement dependency class, which is then used to learn a meta-controller that can be used to explore the various achievements (by leveraging the dependency relationships between the achievements). The authors also argue for the fact that these structures once learned could be used across various instances that share the same achievement structures.


**Summary Of The Review:**

Even with all the reservations I mentioned above, in general I am leaning towards accepting the paper. However, I would definitely urge the authors to see if they could include additional experiments that compare against baselines that can actually leverage the traces in some meaningful way (using an IRL-based method for reward shaping signal could be one such way).

---

> ### Author Response · Authors · 2022-11-16
> **Author Response to Reviewer kuVW**
>
> We thank the reviewer for the useful feedback and insights.
>
> ### For collecting trajectories
>
> > this is built on the assumption that there exists a small/easy enough instance which shares the same structure, but where the exploration can be performed effectively (or where the expert can give the traces)
>
> The reviewer argues that our algorithm may fail when there is no small enough instance with a shared structure, that can be learned. We agree with the reviewer's observation, but argue that having a shared structure is not a hard requirement for our algorithm to work. Even if the solvable achievements are pairwise independent, the exploration still benefits from our algorithm since we can train sub-policies that potentially reliably unlock each achievement and start exploration from there, while excluding distracting reward signals from other achievements.
>
> > if a set of trajectories is available, a possibility that should have been considered was whether you could have used an inverse reinforcement learning algorithm to learn a potentially informative reward function.
>
> We thank the reviewer for the inverse RL suggestion. We would like to experiment it but we fail to find an inverse RL implementation that supports image input and runs fast enough in the limited rebuttal period. We conjecture that inverse RL mostly helps with learning the already discovered achievements, which the vanilla IMPALA baseline can already do, rather than exploring new achievements. We hope to include inverse RL in future updates.
>
> > One possibility the authors might want to consider is, how easy would be for domain experts to directly specify, at least an incomplete version of the achievement dependency graph.
>
> We agree with the reviewer. In our work, we focus on not using domain experts. However, we appreciate the reviewer providing this insight, and hopefully we will address this scenario in our follow-up works.

---

### Official Review · Reviewer_uoYd · 2022-10-25

**Confidence:** 4
**Correctness:** 4
**Technical Novelty And Significance:** 4
**Empirical Novelty And Significance:** 4
**Recommendation:** 8

**Clarity, Quality, Novelty And Reproducibility:**

The paper is very well written and, it is thorough in its exposition and provides clear and compelling results. It is also novel, however it lacks in reproducibility and could improve from clarifying a few points:
Can the authors clarify what they mean by “removing node i-j can only move node j” to the left “ should it be LEFTMOST(G,j)?
How can the non-causal relations break the exploration?
In the large data regime does having a structural model help or hinder results if the model is not well estimated? Can the authors provide an intuition or ablation to understand the trade-offs of using a structured model.
Can the authors provide an intuition of how to properly select the number of clusters, is this sensitive to the number of clusters? How so?
What is the empirical benefit of using the det penalization?

Minor comments/typos:
What is y in eqr_ in page 4
Then use the build -> then build page 2

**Strength And Weaknesses:**

Strengths:
This paper introduces to the best of my knowledge a structured exploration strategy based on hard clustering of achievements and uses its learned graph representation to make meaningful exploration. The idea is novel and relevant to the exploration community.
Weaknesses:
The main weakness of this paper is reproducibility since little information is provided regarding the meta controller policies learned (what model is used/architecture choices/ how it was trained/etc).The authors claim the meta controller is biased towards continuing achievements but how exactly is this biasing being done?
How are the nodes related to the clusters 1-1? how are the edges being estimated are they weighted?


**Summary Of The Paper:**

This paper proposes an structured exploration approach by learning the structure of the achievement task in procedural generated environments and using that structure to explicitly control for which achievement to pursue and learning an achievement conditional policies. The paper shows promising results in Crafter and TreeMaze.


**Summary Of The Review:**

This paper provides a meaningful contribution to the exploration community it provides interesting results in procedurally generated environments. The paper could benefit from further clarification on the above points.

---

> ### Author Response · Authors · 2022-11-16
> **Author Response to Reviewer uoYd**
>
> We thank the reviewer for the positive feedback. We have fixed the typos and included the details of the meta-controller in Section C.1 in the appendix in our updated paper.
>
> > Can the authors clarify what they mean by “removing node i-j can only move node j” to the left “ should it be LEFTMOST(G,j)?
>
> Yes. This sentence means removing the edge i->j can only let node j appear earlier in the topological order, i.e. reduce LEFTMOST(G,j).
>
> > How can the non-causal relations break the exploration?
>
> Non-causal relations won't *break* the exploration! They add unnecessary restrictions to the exploration which makes it much harder if not impossible. Our recovered graph that may include non-causal relations is still valid, meaning every achievement are reachable following this graph. As shown in the "sub-policy learning and exploration" part in Section 5.4, SEA is able to explore and master new achievements (the hard achievement set, see Table 2) despite using an imperfect dependency graph with non-causal relations.
>
>
> > Can the authors provide an intuition or ablation to understand the trade-offs of using a structured model.
>
> Firstly, the structured meta-controller ensures that when learning each individual sub-policy, all the prerequisite achievements have been unlocked, making the learning much easier. Secondly, from the perspective of *exploration for new achievements*, the meta-controller can remove the distracting reward signals that may prevent the agent from finding new tasks. An example of it is in our Crafter experiments, after collecting a stone, the vanilla agent fails to craft stone tools with the stone. This is because dropping the stone unlocks the "place_stone" achievement and grants a reward of 1. The greedy agent always opts to immediately drop the stone after collecting it for the "place_stone" reward, leaving no materials for the subsequent tasks. Our algorithm is able to filter out the "place_stone" reward when exploring from the "collect_stone" achievement, making crafting stone tools easier to learn.
>
> For ablations, removing the structure entirely from our algorithm results in the vanilla IMPALA algorithm, whose performance comparison can be found in Table 2. Ablation with a randomized dependency graph can be found in Figure 7.
>
> > Can the authors provide an intuition of how to properly select the number of clusters, is this sensitive to the number of clusters? How so?
>
> There are multiple ways to automatically select the number of clusters (for a reference, see [here](https://towardsdatascience.com/an-approach-for-choosing-number-of-clusters-for-k-means-c28e614ecb2c)). In our experiments, we use the Silhouette method but we conjecture that any clustering method could work since the representations learned using our method yield easily separable achievement clusters. Our algorithm can be potentially sensitive to the number of clusters. However, even if the number of clusters is wrong, as long as a subset of the achievements is correctly clustered, our algorithm can still use that subset of achievements and the corresponding structure to improve the exploration. To do so, we can discard the clusters with low confidence and build the dependency graph with the remaining clusters.
>
> > What is the empirical benefit of using the det penalization?
>
> We add an additional ablation experiment using average distance penalization instead of det penalization on Crafter. The results can be found in Table 1. Without the det penalization, the clustering works slightly worse in Crafter but completely fails in TreeMaze-Hard.
>
>
> > In the large data regime does having a structural model help or hinder results if the model is not well estimated?
>
> It's possible that having a not well-estimated achievement representation model can hinder the results.
> While this work does not explore this direction, we hope to address this problem in the follow-up work.

---

> > ### Comment · Reviewer_uoYd · 2022-11-17
> > **thank you for the comments**
> >
> > I would like to thank the authors for the clarifications, most of my concerns have been addressed.

---

### Official Review · Reviewer_Jys6 · 2022-10-25

**Confidence:** 4
**Correctness:** 3
**Technical Novelty And Significance:** 3
**Empirical Novelty And Significance:** 2
**Recommendation:** 5

**Clarity, Quality, Novelty And Reproducibility:**

* The paper is generally clearly written.

* The paper appears to be missing several key details, citations, and comparisons in evaluation, so the quality is questionable.

* The paper’s novelty is not clear given the missing references and corresponding discussion.

* The paper is reproducible because the code is provided. However, a number of important details appear to be missing from the document.

**Strength And Weaknesses:**

Strengths:

* The motivation is good, and the method is very intuitive given the problem statement.

* The paper is generally well-written and the method is clearly explained.

* The chosen evaluation environments are appropriate given the motivation, and the method appears to significantly outperform the baselines, especially on the more challenging (longer-horizon) tasks.

Weaknesses/questions:

* A number of crucial details are missing (and are not to be found even in the Appendix). For example, the authors state they “use a controller that topologically traverses the dependency graph. Specifically, at the beginning of the episode or upon completion of any achievement, the controller randomly selects a unfinished achievement whose dependent achievements have all been completed. In our implementation, we also bias the selection probability towards a direct child of the last completed achievement to provide continuity.” How is this bias implemented/parameterized? Another example is that $\phi$ does not appear to be discussed in detail; i.e. how it is implemented/parameterized.

  * Apart from knowing these kinds of implementation details, it would also be useful to see how robust the method is to them in the evaluation section (e.g. varying the hyperparameter controlling how the selection probability is biased).

* There are a number of crucial missing related works that should be at the very least be discussed, and potentially included in the evaluation section:

  * [1] This is a highly relevant work with a very similar setting, motivation, and evaluation environments. They use environment “milestone signals” (similar to sparse rewards in this work) to learn an an “affordance classifier” (like the achievement classifier in this work) to improve exploration at the meta-controller level. Even a similar contrastively-learned representation is used, which they dub an “achievement context” (like the achievement representation in this work). Given the striking similarities, adequate discussion of [1] is crucial to understanding the contributions of the current work, and the comparative limitations/benefits.

  * [2] This is just one example of another relevant work that similarly autonomously learns a subtask dependency structure from experience and utilizes it for better performance. The current work lacks comparison to these types of relevant works in the literature. How does the present approach compare to other works that have also thought about subtask dependency structures? What are the limitations/benefits of the present work in comparison?

  * [3] This work is cited, but improperly so—-the authors in the same sentence mention that reward machines must be hand-specified, but this work learns them from experience. A clearer/more faithful explanation of the differences to the present work should be included.

* The method diagram was more confusing than helpful to me; for example:

  * “Filtered reward” is a phrase that appears nowhere in the text, but it seems to indicate that only achievement-specific rewards are used (as determined by the achievement classifier) to train each subpolicy.

  * Why is the goal/achievement being passed to the achievement classifier in the diagram? My understanding is that the achievement classifier takes in a transition and outputs which achievements are afforded.

  * Minor, but the term “goal” is used in the diagram, which is inconsistent with the terminology in the paper (should be “achievement”).

* In the evaluation there are no comparisons to other HRL approaches, much less other approaches that take advantage of achievement/subtask structure (e.g. [1]). This makes the results less compelling. Without the proper comparisons, it is difficult to judge the contributions of the present work. For example, a simple baseline/ablation would be to perform the clustering to identify achievements, and then use the standard random exploration in the meta-controller, and compare this to the full approach which uses the inferred dependency graph to guide exploration. This would help disentangle the factors leading to performance gains on the evaluation environments.



[1] Possibility Before Utility: Learning And Using Hierarchical Affordances (Costales et al., 2022)

[2] Meta reinforcement learning with autonomous inference of subtask dependencies (Sohn et al., 2020)

[3] Learning Reward Machines for Partially Observable Reinforcement Learning (Icarte et al., 2019)



**Summary Of The Paper:**

This paper proposes a method for learning the “achievement” (i.e. subtask) structure of an environment from offline data, with sparse rewards indicating when achievements are attained. First, an achievement representation is learned using a contrastive-learning type procedure over the transitions where achievements were attained in the offline data. Next, using an achievement classifier constructed from the learned representation, a dependency structure of achievements is built, represented as a DAG. This dependency structure is used to inform more effective exploration over achievements, as opposed to the default random exploration of the meta-controller. The authors demonstrate the efficacy of their approach on tasks within two achievement-oriented environments, relative to baselines that do not take advantage of the achievement structure.

**Summary Of The Review:**

The work is well-motivated and generally clearly written, but is missing key details, citations and corresponding discussion, and comparisons/ablations in the evaluation.

---

> ### Author Response · Authors · 2022-11-16
> **Author Response to Reviewer Jys6**
>
> We thank the reviewer for the helpful feedback.
>
> ### Additional details \& Clarification
>
> > Missing details
>
> - We've included additional details of the meta-controller in Section C.1 in the appendix.
> - We've included additional details regarding the implementation of $\phi$ in Section C.4 in the appendix.
>
> > Missing related works \& HRL baseline
>
> We appreciate the reviewer pointing out the missing related works. The algorithm Hierarchical Affordances Learning proposed in [1] has been added as a baseline in our paper update (see Table 2).
> **We request the reviewer to refer to the [common response for a high-level comparison](https://openreview.net/forum?id=NDWl9qcUpvy&noteId=SyNOgAhRbS) between our work and HRL prior work.**
>
> The additional detailed comparisons are as follows.
>
> - The *affordance representation* in [1] is not the same concept as the *achievement representation* in our work. The *affordance representation* is used to learn the affordances of the subtasks, i.e. which tasks can be achieved at a particular state. In our work, the *achievement representation* is used to cluster and classify the achievements of the environment. Apart from the affordance learning part, [1] falls into the normal hDQN category, whose differences from our work are discussed in the common response.
> - [1] and [2] are similar in that they require the environment to provide affordance signals of the subtasks, in addition to the individual subtask completion signals as typical hDQN-based algorithm need, both of which our algorithm doesn't need. We argue that affordance signals are extremely useful in exploring new tasks, which is a major result of our paper.
> - For [3], we acknowledge that our statement that it requires hand-specified reward machines is improper. The reward machines in [3] is indeed learned but the labeling function $L$ is provided by the environment, which is not assumed by our work. Moreover, the reward machine in [3] scales with $2^{\mathcal{P}}$, where $\mathcal{P}$ in our context is the achievement set, which makes it intractable since we are dealing with up to 30 achievements (21 in Crafter).
>
> > Clarification on The method diagram
>
> We have now updated the method diagram and added text descriptions to better understand the diagram.
> The original figure illustrated that the achievement classifier inputs the target achievements proposed by the meta-controller and filters out the reward signal for that achievement. This has now been updated to be less cluttered.
> We hope the updated version can provide a more clarified view of our algorithm.
>
>
>
> ### References
>
> - [1] Robby Costales, et al. "Possibility Before Utility: Learning And Using Hierarchical Affordances". ICLR. (2022).
> - [2] Sungryull Sohn, et al. "Meta Reinforcement Learning with Autonomous Inference of Subtask Dependencies". ICLR. (2020).
> - [3] Icarte, Rodrigo Toro et al. “Learning Reward Machines for Partially Observable Reinforcement Learning.” NeurIPS (2019).

---

> > ### Comment · Reviewer_Jys6 · 2022-12-02
> > **Response**
> >
> > I thank the authors for their thorough response and corresponding updates to the text.
> >
> > Many of my concerns have been addressed, but a number still remain. Echoing the sentiment of wFTY, I feel that the contributions may be incremental, and specifically, that the subtask formulation here does not necessarily have clear benefits over previous formulations in the literature. Most importantly, the paper includes insufficient discussion of how they are modeling subtasks in relation to these other works. It is nice to see HAL included as a baseline, but there is little discussion of the differences between these approaches, especially in the assumptions they make/which domains they can be applied to. For instance, HAL (and other works) can allow for multiple instances of the same subtask to be completed within the same trajectory, and for completion to be reversible, whereas SEA is limited to domains where subtask completion is binary and irreversible.
> >
> > I believe the contributions in this work are potentially valuable and could be useful to the research community, but at its current state, the paper does not take enough care to make the appropriate comparisons, so it difficult for a reader to determine the utility of this approach in comparison to others in the literature.

---

> > > ### Author Response · Authors · 2022-12-07
> > > **Author Response**
> > >
> > > > It is nice to see HAL included as a baseline, but there is little discussion of the differences between these approaches, especially in the assumptions they make/which domains they can be applied to.
> > >
> > > For a detailed discussion on how our work differs from HRL works like HAL, we refer the reviewer to the "Comparison with Hierarchical RL" section in our [common response](https://openreview.net/forum?id=NDWl9qcUpvy&noteId=SyNOgAhRbS), which we also include here.
> > >
> > > 1. Our work attempts to **solve a different problem**. Subgoals and subtasks normally refer to the intermediate steps that lead to an ultimate goal/task. Achievements are all equal in nature, although some achievements can be subgoals to other achievements. Our algorithm can automatically determine which achievements can be useful (potentially as subgoals) to other achievements. This lowers the burden for the achievement design, as there is no need to carefully craft a curriculum of intermediate steps that help the agent to learn. Our achievement setting also allows for multiple final tasks that our algorithm can represent and solve in one instantiation.
> > >
> > > 2. We **don't** assume the environment provides achievement/subgoal information, including which achievement/subgoal is completed, how many achievements/subgoals are there in the environment, what achievements can be achieved, what is the ultimate task, etc. For instance, in the HAL [4] work, the agent has access to subgoal completion and affordance signals.
> > > 3. We are learning in more complex domains than the mentioned HRL results. Specifically,
> > >     - The mentioned works are tested in environments that have less than 10 subtasks ([2] features 13 subtasks but the agent can only reliably reach 7 of them while the rest are discarded by the controller; [5] includes more tasks than ours, but they have access to subtask eligibility signals, which is a key piece of information for structure inference and subtask solving that we don't assume having). Our algorithm can reliably reach 20 of the 21 achievements and reach the remaining one with ~4% rate in the Crafter environment.
> > >     - The mentioned works are tested in static environments, as opposed to the procedurally generated reward models in our setting. Finding achievements/subgoals is easier in static environments as the states are similar across multiple runs, making it easier to identify the invariants (the achievements/subgoals).
> > >
> > > > HAL (and other works) can allow for multiple instances of the same subtask to be completed within the same trajectory, and for completion to be reversible, whereas SEA is limited to domains where subtask completion is binary and irreversible.
> > >
> > > In SEA, subtask completion can be completed multiple times and reversible. In fact, in Crafter, most of the tasks can be completed multiple times and are reversible. For example, the agent is allowed to collect wood multiple times, and using the collected wood to craft a table essentially reversed the completion of collecting wood. However, in SEA, we **do not** focus on the completion of the tasks, but their associated achievements instead.
> > >
> > > ## Novelty concerns
> > >
> > > We restate our main contributions and novelties here.
> > >
> > > - We propose a novel algorithm that learns and utilizes the underlying reward structure of the environment to help with RL exploration. This is also pointed out by reviewer uoYd.
> > > - We exploit the achievement concept in RL and develop an algorithm to master the achievement-based environment without requiring any extra information.
> > > - We propose a novel loss function (Eqn. 1) that learns the achievement representation effectively while other baselines fail.
> > > - SEA solves the Crafter environment, unlocking most of the achievements reliably and reaching the hardest achievement with ~4% frequency, outperforming previous baselines on this domain.
> > >
> > >
> > > ## References
> > >
> > > - [1] Kulkarni, Tejas D. et al. “Hierarchical Deep Reinforcement Learning: Integrating Temporal Abstraction and Intrinsic Motivation.” NIPS (2016).
> > > - [2] Lyu, Daoming et al. “SDRL: Interpretable and Data-efficient Deep Reinforcement Learning Leveraging Symbolic Planning.” AAAI (2019).
> > > - [3] Jacob Rafati, et al. "Learning Representations in Model-Free Hierarchical Reinforcement Learning". AAAI. (2019).
> > > - [4] Robby Costales, et al. "Possibility Before Utility: Learning And Using Hierarchical Affordances". ICLR. (2022).
> > > - [5] Sungryull Sohn, et al. "Meta Reinforcement Learning with Autonomous Inference of Subtask Dependencies". ICLR. (2020).

---

### Official Review · Reviewer_wFTY · 2022-10-27

**Confidence:** 4
**Correctness:** 3
**Technical Novelty And Significance:** 2
**Empirical Novelty And Significance:** 2
**Recommendation:** 5

**Clarity, Quality, Novelty And Reproducibility:**

- It feels to me that both the paper writing and clarity could be improved as I mentioned in the main Weaknesses. Regarding the novelty, I feel its contribution is limited.

**Strength And Weaknesses:**

Strengths:
- Exploration in a complex environment with long horizon and high-dimensional input can be challenging for RL. This is an interesting work where it learns achievements representation in the procedurally generated environment, builds the achievement dependency graph based on clustered achievements, and learn sub-policies for each achievement with the meta-controller. It would be intriguing to the community of representation learning in RL.



Weaknesses:
- The presentation of the paper could be improved. For example, what are the differences between the procedurally generated environment and the partially observable environment? Or are they the same? What is an finite internal achievement system? Does this require the task to be well-defined? What's the difference between the setting of achievement-based environments and "sparse and delayed reward signals" setting in [1]? Is the achievements design equivalent to the subgoal/subtask design? What's the unlock rate? How does the learned representation & clustering performance affect the sub-policy learning? There lacks more details for this.
- The literature review should include some of previous work regarding subgoal/subtask learning with Hierarchical RL, such as [1,2,3]. All the work of [1,2,3] divides a complex task into sub-tasks, but they did in different ways. There are some shared intuitions in common between this work and [1,2,3], such as task decomposition, goal-directed policy, meta-controller, etc. In particularly, [3] used unsupervised learning method for learning the representation and there needs a discussion on the differences.
- Its current novelty is somewhat incremental to me.


References:
- [1] Kulkarni, T. D., Narasimhan, K., Saeedi, A., & Tenenbaum, J. (2016). Hierarchical deep reinforcement learning: Integrating temporal abstraction and intrinsic motivation. Advances in neural information processing systems, 29.
- [2] Lyu, D., Yang, F., Liu, B., & Gustafson, S. (2019, July). SDRL: interpretable and data-efficient deep reinforcement learning leveraging symbolic planning. In Proceedings of the AAAI Conference on Artificial Intelligence (Vol. 33, No. 01, pp. 2970-2977).
- [3] Rafati, J., & Noelle, D. C. (2019, July). Learning representations in model-free hierarchical reinforcement learning. In Proceedings of the AAAI Conference on Artificial Intelligence (Vol. 33, No. 01, pp. 10009-10010).

**Summary Of The Paper:**

This paper focuses on the exploration issue for a reinforcement learning (RL) agent in an environment with a long horizon and high-dimensional input, particularly the procedurally generated environment. To address the issue, this work proposes a structured exploration with achievements (SEA) method by discovering the finite and fixed underlying structure in the procedurally generated environment. Finally, experiments were conducted on Crafter and TreeMaze, with results demonstrating its effectiveness in terms of achievement learning, graph recovery, and sub-policy learning and exploration.

**Summary Of The Review:**

According to my comments in both the main Weaknesses and the section of Clarity, Quality, Novelty, I feel this is a paper where reasons to reject outweigh reasons to accept. But I would like to change my score if there is a misunderstanding.

---

> ### Author Response · Authors · 2022-11-16
> **Author Response to Reviewer wFTY (1)**
>
> We thank the reviewer for the helpful comments.
>
> ### Concerns on presentation:
>
> We appreciate the reviewer for pointing out issues with comprehension. We have now clarified these problems in our updated manuscript and adding the response here for completeness.
>
> > What are the differences between the procedurally generated environment and the partially observable environment?
>
> *Procedurally generated environment* and *partially observable environment* are two different concepts.
> A *partially observable environment* refers to an environment where the agent doesn't have access to the true environment states, but rather receives an *observation* according to the current state.
> A *procedurally generated environment* refers to an environment that has different rollouts according to different environment initialization seeds. Atari-2600 games is an example of a non-procedural environment, while [Procgen benchmark](https://openai.com/blog/procgen-benchmark/) is a set of procedurally generated environments. Procedurally generated environments are normally harder to explore since the states in different trajectories may differ greatly.
>
> > What is a finite internal achievement system?
>
> A *finite internal achievement system* refers to a set of finite achievements and their corresponding completion identifiers. The reviewer can refer to Section 3.2 in our paper for the formal definition, where $\Gamma$ is the achievement set and $G$ is the completion identifier.
>
> > Does this require the task to be well-defined?
>
> The only assumption that we make about the task is that there exists an identification function that can deterministically determine the completion of that task, given the current state, action, and the next state ($s_t, a_t, s_{t+1}$). The reviewer can refer to the $G$ function in Section 3.2.
>
> > What's the difference between the setting of achievement-based environments and "sparse and delayed reward signals" setting in [1]?
>
> In our setting, *achievement-based environments* have sparse but not delayed reward signals. Our algorithm works in delayed reward signal setting assuming fully-observable states and can be modified to work in partially-observable states by using the recurrent network in our achievement representation model $\phi$. We did not test the delayed reward signal setting since it's not typical and importantly not a key component in our achievement-based environments.
>
> > Is the achievements design equivalent to the subgoal/subtask design?
>
> Subgoals and subtasks normally refer to the intermediate steps that lead to an ultimate goal/task. Achievements are all equal in nature, although some achievements can be subgoals to other achievements. Our algorithm can automatically determine which achievements can be useful (potentially as subgoals) to other achievements. This lowers the standard for the achievement design as there is no need to carefully craft a series of intermediate steps that help the agent to learn. Our achievement setting also allows for multiple ultimate tasks that our algorithm can solve in one go.
>
> > What's the unlock rate?
>
> The unlock rate is the rate of successfully unlocking a certain achievement in multiple runs.
>
> > How does the learned representation & clustering performance affect the sub-policy learning?
>
> Clustering performance would indeed affect the achievement discovery and as a result sub-policy learning. However, in our experiments, baseline clustering algorithms completely failed to cluster the achievements. Hence their performance was closer to random rather than comparable with our method.

---

> > ### Author Response · Authors · 2022-11-16
> > **Author Response to Reviewer wFTY (2)**
> >
> > ### Hierarchical RL comparisons
> >
> > We refer the reviewer to our [common response for the high-level comparison between our work and HRL](https://openreview.net/forum?id=NDWl9qcUpvy&noteId=SyNOgAhRbS) works.
> > In addition to the common response, we also discuss comparisons with papers mentioned by the reviewer. Compared with [3], the additional differences are:
> >
> > 1. The learned subgoals in [3] are states, which won't work in our procedurally generated environment setting as the states hardly overlap across trajectories.
> > 2. The clustering in [3] is done directly on images while ours is done on learned representations. Even if the environment is static, clustering on images is still likely to fail in complex domains where features that are irrelevant to subgoal identification can frequently appear in the images (for example, the health icon in the Crafter environment). Our algorithm deals with this by using the contrast-based learning objective to extract only the relevant features for clustering.
> >
> >
> > ### Concerns on Novelty
> >
> > We thank the reviewer for sharing their opinion. We would like to point out the major contributions of our work.
> >
> >
> > - We propose a novel algorithm that learns and utilizes the underlying reward structure of the environment to help with RL exploration. This is also pointed out by reviewer uoYd.
> > - We exploit the *achievement* concept in RL and develop an algorithm to master the achievement-based environment without requiring any extra information.
> > - We propose a novel loss function (Eqn. 1) that learns the achievement representation effectively while other baselines fail.
> > - SEA solves the Crafter environment, unlocking most of the achievements reliably and reaching the hardest achievement with ~4% frequency, outperforming previous baselines on this domain.
> >
> >
> > ### References
> >
> > - [1] Kulkarni, Tejas D. et al. “Hierarchical Deep Reinforcement Learning: Integrating Temporal Abstraction and Intrinsic Motivation.” NIPS (2016).
> > - [2] Lyu, Daoming et al. “SDRL: Interpretable and Data-efficient Deep Reinforcement Learning Leveraging Symbolic Planning.” AAAI (2019).
> > - [3] Jacob Rafati, et al. "Learning Representations in Model-Free Hierarchical Reinforcement Learning". AAAI. (2019).

---

> > > ### Comment · Reviewer_wFTY · 2022-11-30
> > > **Thanks for the rebuttal**
> > >
> > > Thanks to the authors for the extensive answers and I appreciate your efforts! This is an interesting work focusing on the procedurally generated environment. However, some of my concerns are still not addressed:
> > > - For the procedurally generated environments, my concern is how really different it is from the partially observable environments and whether it is true that all previous algorithms for partially observable environments will fail in this type of environment. According to [4], this type of environment is designed for the robustness of the learned policy, although it's used for testing the abilities of exploration, generalization, reusable skills, credit assignment, and representation as well. But according to Table 2 in the paper, IMPALA and RND outperform SEA on easy sets, which indicates the previous methods don't always fail. To my understanding, there is more stochasticity in the transition dynamics in this type of environment, which makes it more challenging.
> > > - Achievement system: it looks like the achievements depend on the game domain. My question here is how to obtain this completion identifier. Is it pre-defined? or it is provided by the domain as well?
> > > - How does the learned representation & clustering performance affect the sub-policy learning: It would be better if more details can be provided to show how robust the learned representation is.
> > > - Overall, the novelty & contribution seem somewhat incremental to me: First, the achievement structure is similar to the sub-goal/sub-task structure whereas the achievement one comes from the domain side. This limits the generalization ability of the proposed method. Second, the model architecture is a little bit similar to [1, 2, 3] where the pseudo-reward (intrinsic reward) comes from the achievement classifier and graph representation is used for achievement learning. Last, Go-Explore greatly outperforms IMPALA and RND on Montezuma’s Revenge and Pitfall (hard exploration domains), so I am wondering why not select go-explore as a baseline? It would be more convincing if SEA outperforms Go-Explore.
> > >
> > >
> > > References:
> > > - [1] Kulkarni, Tejas D. et al. “Hierarchical Deep Reinforcement Learning: Integrating Temporal Abstraction and Intrinsic Motivation.” NIPS (2016).
> > > - [2] Lyu, Daoming et al. “SDRL: Interpretable and Data-efficient Deep Reinforcement Learning Leveraging Symbolic Planning.” AAAI (2019).
> > > - [3] Jacob Rafati, et al. "Learning Representations in Model-Free Hierarchical Reinforcement Learning". AAAI. (2019).
> > > - [4] Cobbe, K., Hesse, C., Hilton, J., & Schulman, J. (2020). Leveraging procedural generation to benchmark reinforcement learning. In International conference on machine learning (pp. 2048-2056). PMLR.
> > > - [5] Ecoffet, A., Huizinga, J., Lehman, J., Stanley, K. O., & Clune, J. (2019). Go-explore: a new approach for hard-exploration problems. arXiv preprint arXiv:1901.10995.

---

> > > > ### Author Response · Authors · 2022-12-07
> > > > **Author Response**
> > > >
> > > > > how really different it (procedurally generated environments) is from the partially observable environments
> > > >
> > > > **Partially observable** and **procedurally generated** are two independent characterstics of environments. **Partially observable** describes that the agent does not have full access to the state space, but rather has a partial observation of the state. **Procedurally generated** describes the environment is generated by an algorithm each time it resets. A partially observable environment can be either procedurally generated or not, and vice versa. They are not mutually exclusive.
> > > > For example, the Montezuma's Revenge game from the Atari-2600 is a **not-procedurally genarated**, since the environment resets to a same stage with the same distribution of objects every time. On the contrary, the Crafter environment is a **procedurally genarated**. This is because every time the environment resets, an algorithm regenerates a world map with all objects randomly placed. For a visual reference, the reviewer can refer to the [Procgen benchmark website](https://openai.com/blog/procgen-benchmark/). Notice that the Procgen environment generates a completely different stage each time it resets, while in games from Atari-2600, the environment always resets to the same exact scene.
> > > >
> > > > > whether it is true that all previous algorithms for partially observable environments will fail in this type of environment
> > > > > there is more stochasticity in the transition dynamics in this type of environment, which makes it more challenging
> > > >
> > > > We do not make the claim that "all previous algorithms for **partially observable** environments will fail in this type of environment". Our focus is on the **procedurally generated** side, since many previous algorithms are designed for **non-procedurally genarated** environments,, which are static across episodes.
> > > >
> > > > Stochasticity in the transition dynamics is independent of procedural generation. A procedurally generated environment can still have completely deterministic transition dynamics. The challenging part of a procedurally generated environment is that it **resets** to a completely different scene, which makes finding similarities across multiple trajectories difficult.
> > > >
> > > > The Go-Explore algorithm mentioned by the reviewer is a good example. Go-Explore without domain knowledge relies on image down-sampling for state abstraction. In a procedurally generated environment, it would be difficult for two down-sampled image from different trajectories to be the same, given reasonable down-sampling ratio. Even if there are two similar down-sampled images, they are still likely to be not related.
> > > >
> > > > For example, in the Crafter environment, the majority of the image is the surrounding map of the character. Two down-sampled images being the same means the surrounding of the character being similar, which is not a relevant feature for state classification.
> > > >
> > > > > But according to Table 2 in the paper, IMPALA and RND outperform SEA on easy sets, which indicates the previous methods don't always fail
> > > >
> > > > We do not make the claim that previous methods will completely fail to learn! We argue they only fail at hard tasks.
> > > > We emphasize the **exploration** capability of SEA. Specifically SEA is able to learn the **hard tasks**, where all previous method fail. The disadvantage of easy set performance is a trade-off for such exploration capability.
> > > > We would also like to point out that the performance difference of SEA in learning easy set tasks is rather marginal, as >90\% unlock rate is still non-trivial. Importantly, we consider learning new tasks that where previous methods fail. Hence **the large performance gap in hard tasks is more significant algorithmically**, than nominally improving the already sufficient unlock rate on easy tasks.
> > > >
> > > > > Achievement system: it looks like the achievements depend on the game domain. My question here is how to obtain this completion identifier. Is it pre-defined? or it is provided by the domain as well?
> > > >
> > > > The achievement set is designed by the domain. However, we **do not** require access to the completion identifier, nor the achievement set. We only assume that some completion identifier to these achievements exists. Notably, this is in contrast with the hierachical RL works ([1] [2] [3]), where such information is oftenr required. SEA relies on the environment to provide the reward signals, but not which exact achievement has been unlocked. This remains for our algorithm to figure out.

---

> > > > > ### Author Response · Authors · 2022-12-07
> > > > > **Author Response (2)**
> > > > >
> > > > > > How does the learned representation & clustering performance affect the sub-policy learning: It would be better if more details can be provided to show how robust the learned representation is.
> > > > >
> > > > > As we stated in our previous response, baseline clustering algorithms completely failed to cluster the achievements. Hence their performance was closer to random rather than comparable with our method. However, we conduct a synthetic experiment, by adding artificial noise to the achievement classifier, to show how the learned representation & clustering performance affect the sub-policy learning. The result is as follows.
> > > > >
> > > > > |  | Easy set mean | Easy set median | Hard set mean | Hard set median | collect_diamond|
> > > > > | -------- | -------- | -------- | -------- | -------- | -------- |
> > > > > | SEA     | 92.53%     | 96.84%     | 49.30% | 60.70% | 4.21% |
> > > > > | SEA+noise     | 95.38%     | 89.41%     | 21.66% | 20.67% | 0.56% |
> > > > >
> > > > >
> > > > > > First, the achievement structure is similar to the sub-goal/sub-task structure whereas the achievement one comes from the domain side
> > > > >
> > > > > The achievement system in SEA makes a weaker assumption about the domain than the aforementioned HRL research. For a more detailed discussion, we refer the reviewer to the "Comparions with Hierarchical RL" section in our [common response](https://openreview.net/forum?id=NDWl9qcUpvy&noteId=SyNOgAhRbS).
> > > > >
> > > > > > Second, the model architecture is a little bit similar to [1, 2, 3] where the pseudo-reward (intrinsic reward) comes from the achievement classifier and graph representation is used for achievement learning.
> > > > >
> > > > > We do not claim using achievement classifier to filter the intrinsic reward is one of our novelties, as it is a common approach in HRL works. We do consider learning the achievement classifier without domain knowledge to be one of our novelties. Notably, all of [1, 2, 3] use a pre-defined classifier ("internal critic" in [1] and [3], $\mathbb{F}$ in [2]) provided by the environment.
> > > > >
> > > > > **We do not use a graph representation for achievement learning. Instead, we use the result of achievement learning to build a dependency graph**. Furthermore, none of the works in [1, 2, 3] learns a explicit task graph, as done in SEA.
> > > > >
> > > > > > Go-Explore greatly outperforms IMPALA and RND on Montezuma’s Revenge and Pitfall (hard exploration domains), so I am wondering why not select go-explore as a baseline
> > > > >
> > > > > We have discussed insights on why Go-Explore would not be a suitable method in this domain.
> > > > > Go-Explore without domain knowledge relies on image down-sampling for state abstraction. In a procedurally generated environment, it would be difficult for two down-sampled image from different trajectories to be the same, given reasonable down-sampling ratio. Even if there are two same down-sampled images, they still likely to be not related. For example in the Crafter environment, the majority of the image is the surrounding map of the character. Two down-sampled images being the same means the surrounding of the character being similar, which is not a relevant feature for state classification.
> > > > >
> > > > >
> > > > > ## Novelty concerns
> > > > >
> > > > > We restate our main contributions and novelties here.
> > > > >
> > > > > - We propose a novel algorithm that learns and utilizes the underlying reward structure of the environment to help with RL exploration. This is also pointed out by reviewer uoYd.
> > > > > - We exploit the achievement concept in RL and develop an algorithm to master the achievement-based environment without requiring any extra information.
> > > > > - We propose a novel loss function (Eqn. 1) that learns the achievement representation effectively while other baselines fail.
> > > > > - SEA solves the Crafter environment, unlocking most of the achievements reliably and reaching the hardest achievement with ~4% frequency, outperforming previous baselines on this domain.
> > > > >
> > > > >
> > > > > ## References:
> > > > >
> > > > > - [1] Kulkarni, Tejas D. et al. “Hierarchical Deep Reinforcement Learning: Integrating Temporal Abstraction and Intrinsic Motivation.” NIPS (2016).
> > > > > - [2] Lyu, Daoming et al. “SDRL: Interpretable and Data-efficient Deep Reinforcement Learning Leveraging Symbolic Planning.” AAAI (2019).
> > > > > - [3] Jacob Rafati, et al. "Learning Representations in Model-Free Hierarchical Reinforcement Learning". AAAI. (2019).

---

### Author Response · Authors · 2022-11-16
**Common Response**

We thank the reviewers for their helpful feedback and constructive suggestions.

We are glad that the reviewers find out paper "well written" (Jys6, uoYd, kuVW), "clearly explained" (Jys6), "easy to follow" (kuVW), and "thorough in its exposition" (uoYd); our method "intriguing to the community of representation learning in RL" (wFTY), "interesting" (wFTY), "very intuitive" (Jys6), "novel and relevant to the exploration community" (uoYd); our experiment settings "appropriate given the motivation" (Jys6); from the experiment results, our method "demonstrating its effectiveness in terms of achievement learning, graph recovery, and sub-policy learning and exploration" (wFTY), "significantly outperform the baselines, especially on the more challenging (longer-horizon) tasks" (Jys6), and "provides clear and compelling results" (uoYd). Furthermore, reviewer uoYd finds our paper "provides a meaningful contribution to the exploration community" and reviewer kuVW deems our paper "serves well as a starting point".

### Summary of changes

In light of the reviewers' feedback, we have added two more experiments to our paper.
- HAL[4] has been added as an HRL baseline in Section 5.4, suggested by reviewer wFTY and Jys6.
- We have also added a discussion on comparison with HRL.
- A new ablation experiment testing the effectiveness of det penalization has been added to Section 5.2, suggested by reviewer uoYd.
- We have also updated Figure 2 and Figure 8.
- We have updated our paper to reflect the changes with the text updates marked in red color.




### Comparison with Hierarchical RL

Reviewers noted that this work has similarities with HRL and a comparison would be valuable ([1, 2, 3, 4, 5]). We have now expanded the discussion in the paper (and appendix) to include HRL. **Summarily, our method outperforms HRL despite accessing less information than HRL**.

Notably, despite conceptual similarity with HRL, our work still has major differences, namely:

1. Our work attempts to solve a different problem. Subgoals and subtasks normally refer to the intermediate steps that lead to an ultimate goal/task. Achievements are all equal in nature, although some achievements can be subgoals to other achievements. Our algorithm can automatically determine which achievements can be useful (potentially as subgoals) to other achievements. This lowers the standard for the achievement design as there is no need to carefully craft a series of intermediate steps that help the agent to learn. Our achievement setting also allows for multiple ultimate tasks that our algorithm can solve in one go.

2. We don't assume the environment provides achievement/subgoal information, including which achievement/subgoal is completed, how many achievements/subgoals are there in the environment, what achievements can be achieved, what is the ultimate task, etc. For instance, in the HAL [4] work, the agent has access to subgoal completion and affordance signals.

3. We are learning in more complex domains than the mentioned HRL results. Specifically,
    - The mentioned works are tested in environments that have less than 10 subtasks ([2] features 13 subtasks but the agent can only reliably reach 7 of them while the rest are discarded by the controller; [5] includes more tasks than ours, but they have access to subtask eligibility signals, which is a key piece of information for structure inference and subtask solving that we don't assume having).  **Our algorithm can reliably reach **20 of the 21 achievements** and reach the remaining one with ~4% rate in the Crafter environment**.
    - The mentioned works are tested in static environments, as opposed to the procedurally generated reward models in our setting. Finding achievements/subgoals is easier in static environments as the states are similar across multiple runs, making it easier to identify the invariants (the achievements/subgoals).

Additionally, we choose HAL (proposed in [4]) as a representative HRL algorithm as a new baseline in our paper update. The readers can refer to Table 2 for the result. It is worth noting that HAL has access to additional achievement information, including individual completion signals and affordances. Despite having this advantage over our algorithm, HAL still underperforms our method by a substantial margin.

### References

- [1] Kulkarni, Tejas D. et al. “Hierarchical Deep Reinforcement Learning: Integrating Temporal Abstraction and Intrinsic Motivation.” NIPS (2016).
- [2] Lyu, Daoming et al. “SDRL: Interpretable and Data-efficient Deep Reinforcement Learning Leveraging Symbolic Planning.” AAAI (2019).
- [3] Jacob Rafati, et al. "Learning Representations in Model-Free Hierarchical Reinforcement Learning". AAAI. (2019).
- [4] Robby Costales, et al. "Possibility Before Utility: Learning And Using Hierarchical Affordances". ICLR. (2022).
- [5] Sungryull Sohn, et al. "Meta Reinforcement Learning with Autonomous Inference of Subtask Dependencies". ICLR. (2020).

---

### Author Response · Authors · 2022-11-17
**Reminder to the Reviewers**

Dear Reviewers,

Thank you for providing insightful feedback to help us improve our work. We have addressed all the questions in detail and revised the manuscript. Please have a look at our response to each question and the revised manuscript. Please let us know if there is anything that needs further clarification. Thank you.

Best Regards, Authors

---

### Decision · Program_Chairs · 2023-01-20

**Decision:**

Accept: poster

**Justification For Why Not Higher Score:**

The paper presents a promising approach and solid empirical validation. At the same time, its clarity and placement in the context of related work could be improved further. Given these points, acceptance as a spotlight does not seem warrented.

**Justification For Why Not Lower Score:**

Among the reviewers, two strong supporters of the paper make valid points about the novelty and sizable contribution made by the paper.

**Metareview: Summary, Strengths And Weaknesses:**

The paper addresses the problem of hard exploration in complex environments with an internal achievement system. The authors propose a novel approach, Structured Exploration with Achievements (SEA), which learns representations of achievements from offline data. SEA is evaluated on procedurally generated environments.

Reviewers initially noted that the paper addresses the very challenging problem of exploration in problems with long horizons and high-dimensional observations in a novel and well-motivated approach. Empirical results were performed on difficult domains that are well matched with the challenges captured in the problem formulation and addressed by the proposed approach.

Reviews noted several open questions and concerns, many of which were addressed during the rebuttal phase. Clarity needed to be improved, e.g., to provide details about the achievement system and the resulting underlying assumptions, as well as aspects of the approach, e.g., the mechanism behind biasing the meta-controller. Questions were raised regarding robustness to hyperparameter choices, and an ablation of algorithmic components was suggested. Finally reviewers noted the need to better place the work in the context of prior work in related areas, in particular hierarchical reinforcement learning (HRL).

Authors addressed many of the concerns. In particular, they added a novel baseline (an adapted version of HAL, an HRL approach) and new ablation study. They provided additional discussion on prior work in the context of HRL methods, clarified aspects of the approach and provided additional experiment details.

Reviewers provided feedback and additional discussion, both with reviewers and internally. While no full consensus among reviewers has been reached, two reviewers are in strong support of accepting the paper based on its novelty and strong empirical results.

There are some remaining concerns. In particular, the authors are encouraged to further improve the contextualization of their notion of "achievement structure" and how it differs from structural assumptions made in prior work, such as HRL. It is important that readers appreciate where the present work makes, e.g., weaker assumptions than prior work.

The recommendation is to accept the work based on its novelty and strong empirical contribution. Authors are encouraged to carefully consider all reviewer suggestions in their camera ready version, especially to identify ways to further contextualize the work in the context of prior problem formulations and approaches.

**Note From Pc:**

if the above contains the word "oral" or "spotlight" please see: "oral" presentation means -> notable-top-5% and "spotlight" means -> notable-top-25%. As stated in our emails, we are disassociating presentation type from AC recommendations